# Development of a Protein Scaffold for Arginine Sensing Generated through the Dissection of the Arginine-Binding Protein from *Thermotoga maritima*

**DOI:** 10.3390/ijms21207503

**Published:** 2020-10-12

**Authors:** Giovanni Smaldone, Alessia Ruggiero, Nicole Balasco, Luigi Vitagliano

**Affiliations:** 1IRCCS SDN, Via Emanuele Gianturco, 113 80143 Naples, Italy; 2Institute of Biostructures and Bioimaging, CNR, Via Mezzocannone 16. I-80134 Naples, Italy; nicole.balasco@unina.it (N.B.); luigi.vitagliano@unina.it (L.V.)

**Keywords:** protein scaffolds, argininemia diagnosis, biosensors, crystal structure, protein dissection, protein stability

## Abstract

Arginine is one of the most important nutrients of living organisms as it plays a major role in important biological pathways. However, the accumulation of arginine as consequence of metabolic defects causes hyperargininemia, an autosomal recessive disorder. Therefore, the efficient detection of the arginine is a field of relevant biomedical/biotechnological interest. Here, we developed protein variants suitable for arginine sensing by mutating and dissecting the multimeric and multidomain structure of *Thermotoga maritima* arginine-binding protein (TmArgBP). Indeed, previous studies have shown that TmArgBP domain-swapped structure can be manipulated to generate simplified monomeric and single domain scaffolds. On both these stable scaffolds, to measure tryptophan fluorescence variations associated with the arginine binding, a Phe residue of the ligand binding pocket was mutated to Trp. Upon arginine binding, both mutants displayed a clear variation of the Trp fluorescence. Notably, the single domain scaffold variant exhibited a good affinity (~3 µM) for the ligand. Moreover, the arginine binding to this variant could be easily reverted under very mild conditions. Atomic-level data on the recognition process between the scaffold and the arginine were obtained through the determination of the crystal structure of the adduct. Collectively, present data indicate that TmArgBP scaffolds represent promising candidates for developing arginine biosensors.

## 1. Introduction

l-arginine, thereafter denoted as arginine, is a fundamental nutrient of living organisms [1]. Indeed, this amino acid, which is semi-essential for mammals, is required in protein synthesis and in other important metabolic pathways. Arg is also a precursor for the synthesis of nitric oxide (NO), making it important in the regulation of blood pressure and in tumorigenesis [2,3,4,5]. However, arginine metabolism is complex, as multiple enzymes utilize this metabolite as substrate [6].

The accumulation of arginine in the human body as a consequence of defects in the arginase I enzyme causes hyperargininemia (OMIN 207800), an autosomal recessive disorder [7]. Alterations of enzymes linked to arginine metabolism have been recently implicated in Alzheimer disease (AD) [8].

In this scenario, the efficient detection and quantification of the arginine is a field of relevant biomedical/biotechnological interest. Natural arginine binders are the evident candidates for developing this type of biosensors. Among these, substrate-binding proteins (SBP) that deliver arginine and other metabolites to the ABC cassette system for their transportation across the periplasmatic membrane appear to be particularly appropriate for their specificity and stability. In the last few years, we have characterized the arginine-binding protein isolated from the hyperthermophilic organism *Thermotoga maritima* (TmArgBP) [9,10,11,12,13]. This protein presents a number of distinctive/unique features in the large class of SBP. Although substrate-binding proteins are generally monomeric, TmArgBP is essentially dimeric, with the concomitant presence, as minor components, of higher oligomerization states [14,15,16]. The crystallographic structure of the protein has highlighted that its dimer is stabilized by swapping of the terminal *C*-terminal helix [9] (Figure 1). The three-dimensional structure of the protein has also shown that the arginine binding does not only produce a remarkable tertiary structure variation of TmArgBP but also a radical rearrangement of its quaternary structure. The biophysical characterization of the protein has also revealed that it is endowed with an unusual stability against temperature, chemical denaturants and pressure [12,17,18,19]. We have exploited the extraordinary stability of TmArgBP to effectively manipulate the protein through mutation, truncation and dissection of its native structure [10,13,14]. In particular, we have shown that a stable monomeric form may be obtained either though site-directed mutagenesis [15] or through the deletion of the *C*-terminal end, the swapping element in the protein binding [10]. TmArgBP can be further manipulated by dissecting it into its constitutive D1 and D2 domains (Figure 1).

Taking into account these properties, TmArgBP appears a promising system for developing an arginine biosensor. Unfortunately, no fluorescence signal is emitted by the protein due to the lack of Trp residues close to the arginine-binding site. Indeed, the only Trp present in the sequence of the protein (W243) is located at the *C*-terminal end of the protein far from the arginine binding pocket (Figure 1). Therefore, any attempt to devise a TmArgBP-based sensor must proceed through modifications of the wild-type protein. Interesting variants of the protein have been hitherto developed by using as a scaffold the TmArgBP dimer [16,20]. Taking into account the relative inhomogeneity of the wild-type TmArgBP that presents trimeric and higher oligomeric forms along with the dimeric state and the virtually irreversible arginine binding to the protein, we here attempted to generate novel TmArgBP-derived scaffolds using simplified variants. In particular, we here exploited the possibility to manipulate this protein to generate potential arginine sensors. Through combined dissections and mutations of TmArgBP, we were able to generate distinct protein scaffolds whose tryptophan fluorescence was sensitive to the arginine binding (Appendix A). In more general terms, our data also demonstrate that extremely stable substrate-binding proteins, which are very promising systems for developing biosensors, may be effectively manipulated to obtain simplified scaffolds that also reversibly bind the ligand.

## 2. Results

### 2.1. Design of a Potentially Fluorescent Mutant and its Characterization by Computational Techniques

The inspection of the binding pocket of TmArgBP indicates that the protein anchors the aminoacid through a network of different types of interactions. These include hydrophobic interactions involving both aliphatic and aromatic residues, hydrogen bonds and salt bridges that are deputed to the binding of the three charged terminal groups of arginine. Of particular relevance is the interaction that the guanidinium moiety of the ligand makes with the side chains of Phe38 and Phe76 (Appendix A) [21]. Based on these observations, we identified Phe76 as a promising candidate to be replaced with a Trp residue since its environment is highly depending on the binding state of the protein. To this aim, by using computational techniques, we preliminary investigated the impact of a Phe→Trp mutagenesis on this site. In principle, the replacement of the Phe side chain with the bulkier indole group of Trp could induce some local clashes that could limit or even prevent the binding of the ligand. Since we were interested in designing simplified TmArgBP variants, we used the structure of the truncated monomeric variant TmArgBP^20−233^ as a scaffold. Initially, we manually modelled the Trp side chain inside the binding pocket by exploring the most frequent rotameric states of the residue [22]. Although most of these rotamers could not be allocated into the binding pocket without major steric clashes, the one characterized by χ^1^ and χ^2^ dihedral angles of 178° and 105° did not produce any local strain within the protein and presumably did not hamper the arginine binding (Appendix A).

In order to corroborate this preliminary observation, we performed a molecular dynamics (MD) simulation on the F76W variant of TmArgBP^20−233^ (TmArgBP^20−233_F76W^) in its arginine-bound state. The analysis of the structural parameters that are commonly used to evaluate the stability of trajectory structures indicates that TmArgB^P20−233_F76W^ evolves to reach rather stable states in the time interval 20–150 ns (Appendix A). The analysis of the mobility of the W76 throughout the simulation indicates that its side chain assumes a single rotameric state characterized by trans- and gauche- states for χ1 and χ2, respectively (Figure 2A). The visual inspection of the trajectory structures also shows that the arginine is located in the pocket (Figure 2). The analysis of the interactions that the ligand establishes with the protein indicates that it forms stable H-bonds, salt bridges (Appendix A), and hydrophobic interactions (Figure 2E and Appendix A). As shown in Figure 2B,C, the ligand is in close contact with the residue W76 whose environment is expected to change as a function of the protein-binding state. Collectively, these analyses suggest that the mutant TmArgBP^20−233_F76W^ is a suitable candidate to generate a protein variant that exhibits a variation of the Trp fluorescence as a consequence of the arginine binding.

### 2.2. Expression and Characterization of the Mutant TmArgBP^20−233_F76W^

To generate candidates for the development of an arginine biosensor based on simplified variants of TmArgBP, we expressed and characterized the mutant TmArgBP^20−233_F76W^, in which the mutation is inserted in the truncated variant of the protein (see Appendix A for the sequence). The recombinant protein was expressed in high yields and remarkable purity. As expected on the basis of the data available for the parent protein and taking into account the deletion of the swapping element, the mutant is monomeric.

The folding state and the stability of TmArgBP^20−233_F76W^ was evaluated by circular dichroism (CD) spectroscopy. As shown in Figure 3A, the spectrum of the mutant displays the characteristic features shared by proteins adopting an α/β fold. It is also endowed with a remarkable stability as the thermal denaturation curve registered following the CD signal at 222 nm does not present any significant transition (Figure 3B). This observation indicates that, as the parent protein [10], TmArgBP^20−233_F76W^ is stable over the entire range of temperature explored (20–100 °C). To induce the unfolding of this mutant, the increase in the temperature has to be coupled with the addition of 4.0 M GuHCl (Figure 3C). In these conditions, the observed melting temperature is 57 °C.

Since TmArgBP and its variants are produced in arginine-bound forms in the commonly used expression protocols, the ligand was removed using rather harsh conditions that included the presence of the chemical denaturant GuHCl (see Methods for details). The spectrum corresponding to the Trp-fluorescence emission of the unliganded form of TmArgBP^20−233_F76W^ highlights the occurrence of a strong peak centered at 350 nm (Figure 4A). Since the deletion of the C-terminus of TmArgBP also eliminated the only tryptophan residue (W243) present in the wild-type protein, this emission is exclusively due to the tryptophan introduced in the active site upon mutation. To qualitatively and quantitatively measure the effect on the tryptophan fluorescence, TmArgBP^20−233_F76W^ was titrated with increasing amounts of arginine (range 0.5–730 µM). As shown in Figure 4A, the addition of the ligand produced significant effects on the fluorescence spectrum. Indeed, a remarkable reduction in the fluorescence intensity was observed along with a shift of the peak toward shorter wavelengths (~10 nm) in a ligand concentration-dependent manner. Such a shift of fluorescence peaks, which are known to be generated from the tryptophan becoming more buried and shielded away from the hydrophilic environment, represents a strong indication of changes in the local environment of this residue [23]. This variation was quantified by plotting the peak of the tryptophan fluorescence in response to the ligand concentration (Figure 4B). The fitting of the experimental data also provides an estimate of the dissociation constant (K_D_ = 98.3 ± 29 µM) of the adduct formed by the arginine with TmArgBP^20−233_F76W^. Finally, we evaluated the reversibility of the binding by washing the TmArgBP^20−233_F76W^/arginine complex with a Tris buffer solution and dialyzing it for four hours (see Section Materials and Methods for details). Then, the washed sample was titrated with arginine. As shown in Figure 4C, the addition of the ligand did not produce appreciable effects on the fluorescence spectrum. This indicates that the washing procedure adopted could not remove the tightly bound arginine from the protein pocket. This consideration is corroborated by the fact that the value of the wavelength (~335 nm) of the intensity peak observed before the addition of the ligand (Figure 4C) is similar to the one observed for the protein saturated with arginine (Figure 4A).

### 2.3. From the Monomeric to a Single Domain Scaffold

In our efforts to simplify the protein scaffold, we dissected the two-domain structure of the protein into its individual domains (D1 and D2). Although both domains cooperate to the anchoring of the arginine in the wild-type protein (Appendix A), only D1 is able to bind this ligand when considered as isolated entity. In this scenario, we introduced the same Phe→Trp mutation we have previously characterized in TmArgBP^20−233^. Due to the different numbering scheme adopted for TmArgBP D1 (see the notations section in the Materials and Methods), the mutation is F57W and the mutant is denoted thereafter D1^F57W^. As for TmArgBP^20−233_F76W^, a highly pure recombinant form of D1^F57W^ was expressed in high yields.

The CD spectrum indicates that D1^F57W^ assumes the expected α/β fold (Figure 5A). As for TmArgBP^20−233_F76W^, the thermal denaturation curve registered following the CD signal at 222 nm does not present any significant transition (Figure 5B). The melting point of D1^F57W^ was detected at 57 °C when 4.0 M GuHCl was added to the protein solution (Figure 5C).

The tryptophan fluorescence spectrum of this mutant was very similar to that detected for TmArgBP^20−233_F76W^ with a peak of intensity centered at 350 nm (Figure 6A). The intensity and the wavelength of the peak change, in a dose-dependent manner, upon the addition of aliquots of arginine. The fitting of the intensity of the peak collected as function of the arginine concentration yields a k_D_ value of 3.4 ± 0.9 μM (Figure 6B). It is worth mentioning that the dissociation constant observed for D1^F57W^ is significantly lower than that measured for TmArgBP^20−233^. The analysis of the reversibility of the binding of arginine to D1^F57W^ provided quite different results than that obtained for TmArgBP^20−233_F76W^. Indeed, as shown in Figure 6C, the binding of the arginine to this protein scaffold is reversible, as the mutant was able again to bind the ligand after the application of the washing protocol also used for TmArgBP^20−233_F76W^ (see methods for details). Since we have recently shown that the D1 domain of TmArgBP is able to bind the guanidinium ion [17], we also checked whether D1^F57W^ was a suitable candidate for guanidinium sensing. However, as shown in Figure 6D, the addition of this ion did not produce significant variations of the tryptophan fluorescence of the protein. 

### 2.4. Crystal Structure of D1^F57W^

To gain further insights into the properties of the novel TmArgBP variants here designed and characterized here, we undertook crystallographic studies on D1^F57W^. The domain was successfully crystallized and diffraction data were collected at 1.79 Å resolution (Table 1). A three-dimensional model for D1^F57W^ was built on the basis of the electron density maps that were well-defined for the vast majority of the residues. The final model presents good crystallographic indicators and an excellent stereochemistry (Table 1 and Appendix A and Appendix A) that was evaluated using both standard and innovative protocols (see Methods). The asymmetric unit of the crystals contains two copies of the domain whose structure is very similar both at local and at global level (the RMSD value of the two copies is 0.31 Å).

A comparative analysis of the global structure of D1^F57W^ (Figure 7A) with previous structural characterizations of D1 in different binding states or embodied in the native protein indicates that the mutation does not produce major rearrangements (Appendix A). The inspection of the binding pocket (Figure 7 and Appendix A) indicates that the arginine is anchored by a remarkable number of residues that establish a variety of different interactions with the ligand [21]. A remarkable contribution to the recognition of arg is provided by the mutated residue W57 (Figure 7, Appendix A). The values of the side chain dihedral angles (χ^1^; χ^2^) that the residue adopts in the crystallographic structure are (−178.16°, −92.18°) and (174.49°, −108.34°) for the two molecules present in the asymmetric unit. This observation indicates that W57 assumes the rotameric state predicted by the computational analysis reported above. Notably, the accessible surface area of this residue decreases from 104.9 Å^2^, in the absence of the ligand, to 31.6 Å^2^. The drastic change in the local environment of the tryptophan explains the variation of the fluorescence signal associated with the arginine binding. It is worth mentioning that, in addition to the guanidinium moiety, W57 interacts with several other atoms (Appendix A). This observation indicates that almost the entire arg aminoacid contributes to the burying of tryptophan side chain and explains why the binding of the sole guanidinium moiety produces limited effects on the fluorescence emission (Figure 6D).

## 3. Discussion

The detection of arginine is an issue whose importance extends over different scientific areas. In biomedical sciences, the role of this aminoacid goes well beyond its contribution to the building of proteins. Indeed, arginine dysregulations underlie several pathological states [24]. The absence in melanoma and hepatocellular carcinoma of the enzyme argininosuccinate synthase makes the production of arginine unfeasible [25]. Low levels of arginine are found in several cancers, including leukemia and breast cancer, as well as in other clinical states, such as asthma, arthritis and psoriasis [26,27,28,29]. On the other hand, the quantification of arginine in food is an important quality control step, as products of arginine degradation may lead to the formation of carcinogens. In this scenario, it is not surprising that a repertoire of different methods for arginine detection have been developed [30,31]. As insightfully reviewed by Verma et al. [24], these methods, which have been developed in a variety of instrumental conditions, exploit enzymatic essays as well as whole cell and tissue-based tools. Although natural proteins often lack some key requisites, such as specificity and stability, to be used as biosensors there is a growing interest in developing protein-based biosensors. Substrate-binding proteins represent promising candidates to be used as scaffold for biosensors, although substantial engineering may be required. This approach has been applied with some success also for the arginine detection [16,20,32,33]. Here, we exploited the extraordinary stability of TmArgBP to generate novel and simplified scaffolds whose tryptophan fluorescence depends on their arginine binding state. In particular, the possibility to truncate and/or dissect the protein into fragments that are endowed with a remarkable stability against temperature (>90 °C), chemical denaturants, and pressure (10 gigabar) [12,17] prompted us to make Trp fluorescent variants that were sensitive to arginine upload. In this framework, we developed two different scaffolds corresponding to the truncated monomeric form of the protein devoid of its *C*-terminal swapping helix and to the individual D1 domain. In both cases, the replacement of a Phe residue of the binding pocket with a Trp residue was sufficient to detect a clear variation of the Trp fluorescence upon ligand biding while retaining the remarkable stability of the parent protein. It is also worth mentioning that the removal of the *C*-terminal end from these constructs also eliminates the only other Trp residue present in the TmArgBP sequence. Despite the extreme simplicity of D1^F57W^, the individual domain retains a good affinity for arginine (k_D_ of 3 µM). Surprisingly, the larger TmArgBP^20−233_F76W^, in which both D1 and D2 domains cooperate to the anchoring of the arginine, presents a reduced affinity (k_D_ of ~100 µM). Interestingly, the two scaffolds have been designed to exhibit a completely different behavior in relation to the arginine release. Indeed, while TmArgBP^20−233_F76W^, in line with the ligand burying shown by SBP, binds the ligand in an almost irreversible way, D1^F57W^ releases the ligand with a simple washing. It is also worth mentioning that the binding of the guanidinium ion does not produce fluorescence variations. This finding indirectly suggests that small contaminants deriving from arginine degradation should not interfere with the aminoacid quantification. Altogether, we here report the development of two TmArgBP-based different scaffolds for arginine sensing characterized by different sizes and properties. In a more general context, our data indicate that SBP isolated from thermophilic organisms may be effectively engineered to generate miniaturized variants of these proteins that can be used as protein-based biosensors.

## 4. Materials and Methods

### 4.1. Notations

Wild-type TmArgBP is a dimeric protein made of 247 residues (UniProtKB - Q9WZ62). Its *N*-terminal end (residues 1–19) is the signal peptide responsible for the periplasmic localization of the protein. Since this region is highly hydrophobic, it is typically removed in the recombinant forms of the protein. Therefore, the monomeric form of the protein generated through the truncation of the *C*-terminal helix corresponds to the residues 20–233 (TmArgBP^20−233^). In the numbering scheme of the wild-type and of the truncated protein the Phe→Trp mutation characterized here corresponds to residue 76. Since the D1 domain of the protein is made of fragments that are non-consecutive in TmArgBP sequence (residues 20–114 and 207–233), which were artificially connected by a GGGGSG flexible linker, its numbering cannot follow that of the wild-type protein. Therefore, in line with previous studies, the numbering of D1 starts with the residue 1 that corresponds to the residue 20 of the wild-type protein. Consequently, the Phe→Trp mutation here characterized corresponds to residue 57 in the D1 framework.

### 4.2. Molecular Dynamics: Models and Protocol

Fully atomistic Molecular Dynamics (MD) simulations were performed on the arginine bound form of TmArgBP^20−233_F76W^. Atomic coordinates of TmArgBP^20−233^ were extracted from the PDB entry 6GGV. Residue Phe76 was mutated to Trp using COOT (Crystallographic Object-Oriented Toolkit) [34]. The MD simulation was carried out using the GROMACS (GROningen MAchine for Chemical Simulations) software package (version 2019.6) and Amber99SB as force field [35]. The protein model was immersed in a triclinic box (6.042 × 7.806 × 5.755 nm^3^) and solvated with 7637 water molecules of the TIP3P (transferable intermolecular potential with 3 points) model. A minimum distance of 10 Å between the protein and the box was imposed. The system was neutralized adding five sodium counterions. The particle-mesh Ewald (PME) method with a grid spacing of 1.6 Å was used for the electrostatic interactions, whereas a 10 Å cut-off was applied for the Lennard-Jones (LJ) interactions. Bond lengths were constrained with the LINCS (LINear Constraint Solver) algorithm [36]. The system was first energy minimized by means of steepest descent and then equilibrated in two steps: in the first phase, the temperature of the systems was stabilized at 300 K for 500 ps (NVT); equilibration of pressure at 1 atm was then conducted for other 500 ps (NpT). Temperature and pressure control were achieved using the Velocity Rescaling and Parrinello–Rahman algorithms, respectively [37]. We carried out a production run of 150 ns using a time step of 2 fs. The convergence of the simulation was checked by calculating the root mean square inner product (RMSIP) between the two halves of the equilibrated trajectory. To this aim, the motions of the protein C^α^ atoms along the first 10 eigenvectors were considered [38,39]. In detail, the last 130 ns of trajectory was divided in two halves (20–85 ns and 85–150 ns). The high value of the RMSIP (0.77) between these two trajectory portions indicates that a satisfactory level of convergence is obtained. Structural analyses of trajectory frames were carried out with GROMACS routines and the VMD (Visual Molecular Dynamics) molecular visualization program [40].

### 4.3. Cloning, Expression and Purification of the Mutants

The Phe→Trp replacement in the truncated TmArgBP^20−233^ (TmArgBP^20−233_F76W^) and in the D1 (D1^F57W^) was obtained using the QuickChange mutagenesis kit (Agilent) (Appendix A).

The following mutagenic primers were used in this study: F76Fw-5′ GAAGATCGTCGATATGACCTGGGACGGACTCATTCCGAGCC-3′ and F76WRv-5′ GGCTCGGAATGAGTCCGTCCCAGGTCATATCGACGATCTTC-3′. The two mutants were expressed using *Escherichia coli* (E. coli) BL21(DE3) or Rosetta 2 (DE3) cells. Induction was performed at 22 °C for 16 h by the addition of 0.8–0.5 mM Isopropyl β-d-1-thiogalactopyranoside(IPTG). Cells were then harvested and the soluble extract was obtained by sonicating cell pellets in 20 mL of lysis buffer. The purification of TmArgBP^20−233_F76W^ and D1^F57W^ was conducted in a fashion similar to that previously reported [9,11]. To further purify soluble proteins from aggregated forms and/or other contaminants, fractions containing TmArgBP^20−233_F76W^ and D1^F57W^ were loaded on Superdex S75 10/30 pre-equilibrated in a buffer containing 50 mM TrisHCl and 150 mM NaCl (pH 8.0). The ligand-free form of the two proteins was obtained using two different protocols. In the case of TmArgBP^20−233_F76W^, the arginine was removed by dialysis using strong denaturing conditions in the presence of 4M GuHCl [10], whereas, for D1^F57W^, a prolonged dialysis in Tris/NaCl buffer was sufficient.

### 4.4. Circular Dichroism (CD)

CD spectra were registered with a J-810 spectropolarimeter equipped with a Peltier temperature control system (model PTC-423-S, Jasco Europe, Cremella (LC), Italy). Far-ultraviolet measurements (198–260 nm) were performed at 20 °C using a 0.1 cm optical path length cell. The proteins were dissolved in phosphate buffer saline (PBS) buffer (pH 7.4) and tested at 0.2 mg/mL concentration. CD spectra were recorded with a time constant of 4 s, a 2 nm bandwidth, and a scan rate of 10 nm min^−1^. The reported spectra were obtained after averaging the signal over at least three scans. The baseline was corrected by subtracting the complete buffer spectrum. Thermal denaturation curves were recorded over the 20 °C–100 °C temperature range monitoring the CD signal at 222 nm. The curve was registered either by simply dissolving the protein in the PBS buffer or by adding 4.0 M GuHCl to this solution.

### 4.5. Tryptofan Fluorescence Spectroscopy (TFS)

All tryptophan fluorescence spectroscopy experiments were performed using CARY Eclipse Fluorescence Spectrophotometer (Varian Inc. Santa Clara, CA, USA) and a 100 µl quartz cuvette. The excitation wavelength was fixed to 263 nm and emission spectra were collected between 300 and 500 nm with a slit width of 5 nm. The temperature was kept constant at 25 °C. To measure arginine interactions, recombinant proteins were used at a concentration of 45 µM in Tris buffer (pH 8.0) and the fluorescence spectra were reordered with increasing concentrations of ligand from 0 to 730 µM. Control buffer titrations to protein solutions were performed in parallel for background determination in each experiment. Data derived from three independent experiments were analyzed using nonlinear regression with GraphPad Prism 6 by applying the binding model “One Site- Total”. To test the reversibility of the ligand binding, after the analysis, the protein–ligand mixture was rinsed with Tris buffer three/four times using a centrifugal device (Amicon Ultra centrifugal filter, Millipore). After the washing, the arginine binding was tested again by TFS.

### 4.6. Protein Crystallization, Data Collection and Structure Refinement

Crystallization experiments were performed at 293 K on a solution containing the mutated D1 domain by using the hanging-drop vapor diffusion method. Crystals were obtained using a protein concentration of 6 mg mL^−1^ and a medium containing 0.1 M Sodium acetate trihydrate pH 4.5, 25% *w*/*v* Polyethylene glycol 3350. High-resolution diffraction data were collected in house using Cu Kα X-ray radiation (1.54 Å) from a Rigaku Micromax 007 HF generator equipped with a Saturn944 CCD detector at 100 K. The data were collected by flash-cooling in the supercooled N_2_ gas produced by an Oxford Cryo-system after the addition of 20%(*v*/*v*) ethylene glycol to the harvesting solution. Data were processed and scaled using the HKL2000 program package. The crystals, which contain two molecules per asymmetric unit, belong to the P 21 21 21 space group and diffracted at 1.79 Å. Data collection statistics are reported in Table 1.

The structure of the protein was solved by molecular replacement using the program Phaser [41] and the structure of the arginine-bound form of D1 (PDB code 6GPC) as a starting model. A reliable model was then obtained by the automatic modeling performed using ARP/wARP [42]. The crystallographic refinement of the structure has been performed with REFMAC (REFinement of MACromolecular structures) [43]. Since the early stage of the refinement, a clear electron density corresponding to the indole group of the Trp side chain at position 57 was evident in both chains. The electron density is generally of good quality along the entire polypeptide chains. Water molecules were incorporated into the structure in several rounds of refinements. The final model presents R-factor and R-free values of 0.178 and 0.233, respectively. The stereochemistry of the final model was checked by using standard validation protocols such as PDB validator (https://validate-rcsb-2.wwpdb.org/) and Molprobity [44] and innovative approaches that are based on the evaluation of the conformation-dependent variability of the peptide geometry (http://study.ibb.cnr.it/quiproqua/) [45,46,47]. The analysis of the final model carried out with the Molprobity server indicates that it is endowed with a good stereochemistry (Table 1). Indeed, the φ and ψ values of all residues fall in allowed regions of the Ramachandran plot with the vast majority adopting favored conformations (95.7%). Finally, to evaluate the accuracy of bond and dihedral angles of the final model, we checked the variability of the backbone valence angles and of the peptide planarity as a function of the local conformation. As shown in Appendix A and Appendix A, the variability of these parameters follows the trends detected in highly accurate protein structures [45,46,47]. The atomic coordinates of D1^F57W^ have been deposited in the PDB with the identification code 7A99.

## Figures and Tables

**Figure 1 ijms-21-07503-f001:**
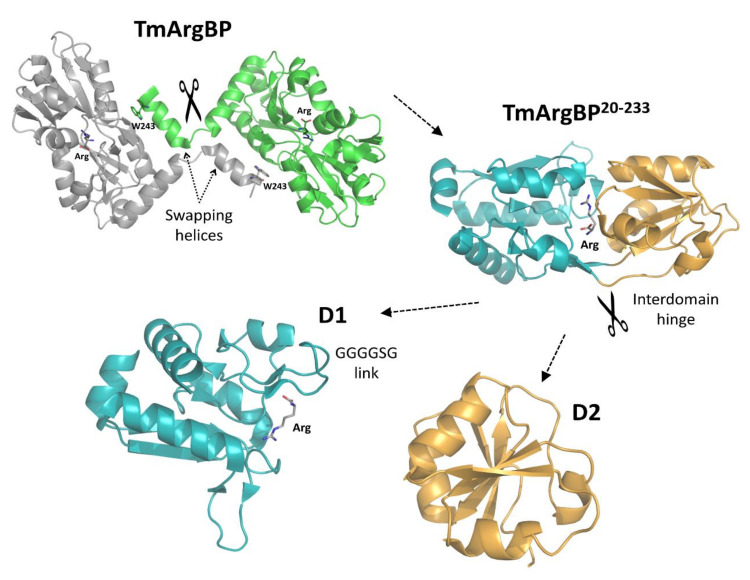
Cartoon representation of *Thermotoga maritima* arginine-binding protein (TmArgBP) truncation and dissection from the wild-type dimer to the truncated monomeric form TmArgBP^20−233^ and, finally, to the individual D1 and D2 domains. The arginine-bound form of TmArgBP (ProteinDataBank ID: 4PSH), of TmArgBP^20−233^ (PDB ID: 6GGV), of D1 (PDB ID: 6GPC) and the D2 domain (PDB ID: 6GPM) have been used to generate the figure. The arginine ligand and residue Trp243 are shown as sticks.

**Figure 2 ijms-21-07503-f002:**
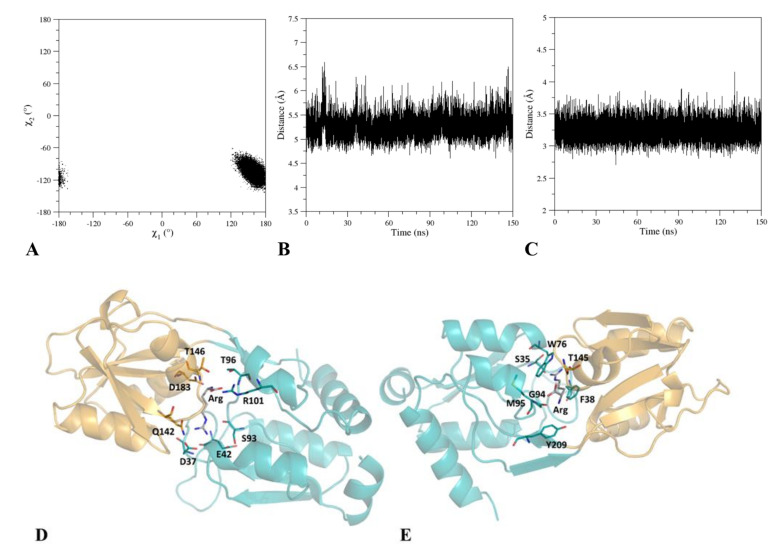
Rotameric states (χ^1^ and χ^2^ dihedral angles) assumed by Trp76 in the molecular dynamics (MD) simulation of TmArgBP^20−233_F76W^ (**A**). Time evolution of the distances between: (**B**) the centers of mass and (**C**) the closest atoms of the arginine ligand and Trp76 residue. Cartoon representation of the arginine environment: residues involved in H-bonding (**D**) or hydrophobic (**E**) interactions with the ligand are shown as sticks. The MD structure (*t* = 77.66 ns) closest to the average structure computed in the equilibrated region of the trajectory (20–150 ns) has been considered.

**Figure 3 ijms-21-07503-f003:**
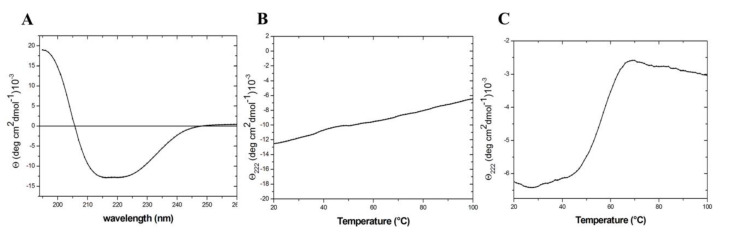
Spectroscopic characterization of TmArgBP^20−233_F76W^: Far-UV circular dichroism (CD) spectrum (**A**), thermal denaturation curves in phosphate buffer saline (PBS) buffer (**B**) and in PBS buffer containing 4.0 M GuHCl (**C**).

**Figure 4 ijms-21-07503-f004:**
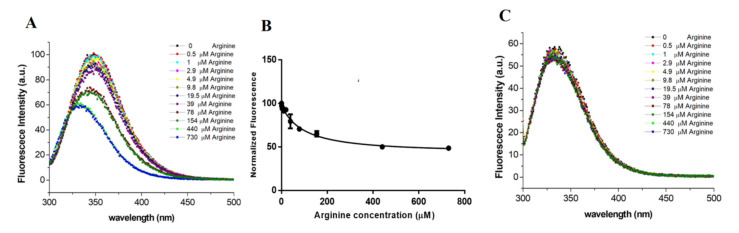
(**A**) Effect of the arginine binding on the Trp fluorescence of TmArgBP^20−233_F76W^. The concentration of the protein was 45 µM. (**B**) Maximum of the fluorescence intensity is reported as a function of the concentration. (**C**) Fluorescence measured on a TmArgBP^20−233_F76W^ sample that was initially saturated with arginine and then extensively washed.

**Figure 5 ijms-21-07503-f005:**
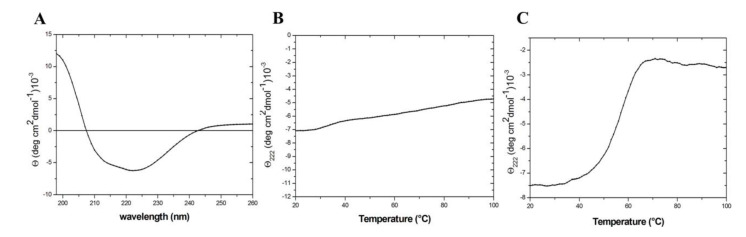
Spectroscopic characterization of D1^F57W^: Far-UV CD spectrum (**A**), thermal denaturation curves in PBS buffer (**B**) and in PBS buffer containing 4.0 M GuHCl (**C**).

**Figure 6 ijms-21-07503-f006:**
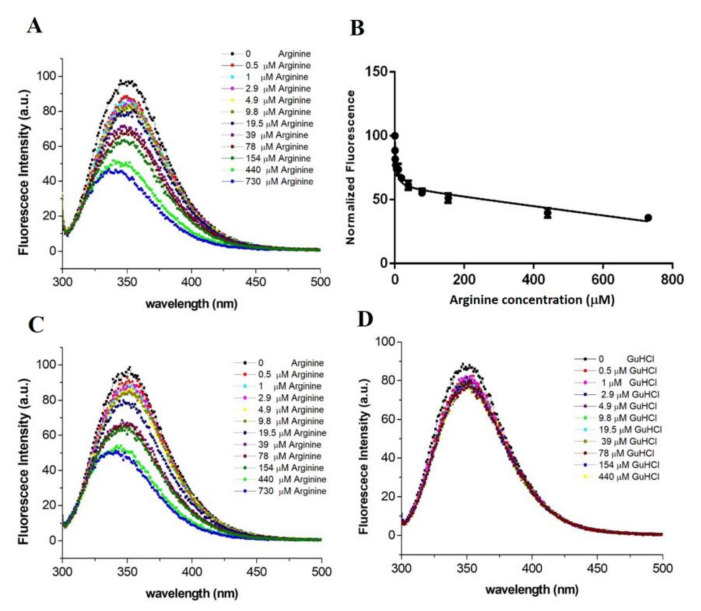
(**A**) Effect of the arginine binding on the Trp fluorescence of D1^F57W^. The concentration of the protein was 45 µM. (**B**) Maximum of the fluorescence intensity reported as function of the concentration. (**C**) Fluorescence measured on a D1^F57W^ sample that was initially saturated with arginine and then extensively washed. (**D**) Variation of the Trp fluorescence of D1^F57W^ upon GuHCl titration.

**Figure 7 ijms-21-07503-f007:**
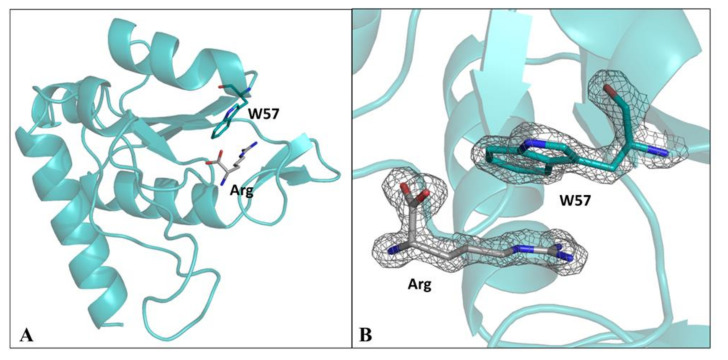
Crystal structure of the arginine-bound D1^F57W^ from TmArgBP. The arginine ligand and W57 residue are shown as sticks (**A**). |2Fo− Fc| electron density map, contoured at 2.0 σ, of the arginine ligand and of the mutated residue W57 (**B**).

**Table 1 ijms-21-07503-t001:** Data collection and refinement statistics of TmArgBP D1^F57W^. Values in parentheses are for the highest-resolution shell (1.87–1.79 Å).

Protein	TmArgBP D1^F57W^
X-ray device	Rigaku FR007HF with CCD detector
Space group	P212121
a, b, c (Å)	37.48, 67.22, 102.41
Resolution range (Å)	50.00–1.79
Wavelength (Å)	1.54
Average redundancy	4.0 (2.6)
Unique reflections	25021 (3013)
Completeness (%)	99.1 (97.7)
R merge (%)	6.1 (34.2)
Average I/σ(I)	27.4 (4.0)
Asymmetric unit	Two molecules
R/R-free	0.178/0.233
No. of atoms	2353
No. of residues	259
No. of water molecules	297
Mean B value (Å^2^)	24.745
R.m.s. bonds (Å)	0.018
R.m.s. angles (°)	1.841
**MolProbity statistics**	
MolProbity score	1.47
Clashscore, all atoms:	3.86
Ramachandran Allowed	257/257
Ramachandran Favored	246/257
Ramachandran Outliers	0/257
Favored rotamers	212/221
Poor rotamers	2/221
Cβ deviation outliers	2/237
Bad bonds	2/2074
Bad angles	1/2795

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
