# Peer review of "Development of a Protein Scaffold for Arginine Sensing Generated through the Dissection of the Arginine-Binding Protein from Thermotoga maritima"

_ijms, 2020, doi:10.3390/ijms21207503_

Round 1

Reviewer 1 Report

This paper reports a study about protein scaffolds suitable for arginine sensing. This study encompasses the suitability of protein scaffolds for arginine sensing, characterizing the properties of the different materials obtained and simulations for their use. It is recommended for publication in the International Journal of Molecular Sciences after major revision indicated below.

GENERAL COMMENTS

  • Improve the quality of the figures.
  • Which is the protein used to the development of the scaffolds. This information is missing, and it is really important concerning the properties of the scaffold. In this line, I suggest including the type of protein used, as well as the characterization of the system prior to arginine joint, (such as FTIR, SEM, etc.) since it would modify the possible connections and interactions.

SPECIFIC COMMENTS

  1. Introduction
  • Include the main novelty of your work.

     Supplementary material:

  • Figures not available.

     References

  • Include more recent references (only 6 out of 43 are from 2018 onwards).
  • Include more references from the journal.
  • Revise reference number 19. The name of the journal is not abbreviated.

Author Response

Comments and Suggestions for Authors

This paper reports a study about protein scaffolds suitable for arginine sensing. This study encompasses the suitability of protein scaffolds for arginine sensing, characterizing the properties of the different materials obtained and simulations for their use. It is recommended for publication in the International Journal of Molecular Sciences after major revision indicated below.

Response

We thank the reviewer for the comment and for the constructive criticisms.

General comments:

Comment #1

  1. Improve the quality of the figures.

Response

As suggested by the reviewer, the quality of the figures 2 and 3 of the original manuscript (now figures 3 and 5) has been improved.

Comment #2

  1. Which is the protein used to the development of the scaffolds. This information is missing, and it is really important concerning the properties of the scaffold. In this line, I suggest including the type of protein used, as well as the characterization of the system prior to arginine joint, (such as FTIR, SEM, etc.) since it would modify the possible connections and interactions.

Response

Taking into account the comment of the reviewer, we reported the amino acid sequences of the developed scaffolds in the revised version of the manuscript (Figure S1). Moreover, we performed additional characterizations of the two variants here generated. In particular, we evaluated their overall structural features and thermal stability using circular dichroism (Figures 3 and 5 of the revised version) as reported in page 4 lines 135-142 and page 11 lines 382-391 of the revised manuscript. Although not surprising, these new experiments demonstrate that these new scaffolds are endowed with a remarkable thermal stability.

Specific comments:

Comment #1

  1. Introduction - Include the main novelty of your work.

Response

The Introduction section has been extended to highlight the main novelties of the present work. In particular, a new paragraph has been added to the revised version of the manuscript in page 3 lines 80-86.

Comment #2

  1. Supplementary material: Figures not available.

Response

We checked on the website that we submitted the supplementary material file embodying the related figures.

Comment #3

References –

(a) Include more recent references (only 6 out of 43 are from 2018 onwards).

(b) Include more references from the journal.

(c) Revise reference number 19. The name of the journal is not abbreviated.

Response

Upon revision we included additional references to the manuscript by following the indications provided by the reviewer. Some recent citations have been added, including some of the journal (ref. number 4, 5, 30 and 31, marked in red.)

Ref. 19 (now ref 21) has been corrected:

Ref 21: “Laskowski, R.A.; Swindells, M.B. LigPlot+: Multiple Ligand-Protein Interaction Diagrams for Drug Discovery. J. Chem. Inf. Model 2011, 51, 2778-2786, doi:10.1021/ci200227u.”

Reviewer 2 Report

Manuscript ID: ijms-943999

Arginine detection is of great interest as alterations in its levels have been associated to a number of human diseases. Smaldone et al. have developed a molecular scaffold that could be used as a promising tool in arginine biosensors. Starting from a simplified version of the Arginine Binding Protein from Thermotoga maritima (TmArgBP) and based on MD simulations, two fluorescent variants are obtained and characterized in terms of 3D structure and fluorescence properties upon arginine binding.

Although the experimental work is well done from a methodological point of view, in my opinion it is not enough to reach the caliber of ijms:

In their 2018 International Journal of Biological Macromolecules paper (“Domain communication in Thermotoga maritima Arginine Binding Protein unraveled through protein dissection” 119, 758-709), the authors described the dissection of TmArgBP into its monomeric version and its D1 and D2 domains (Fig 16 in this paper is basically the same as Figure 1 in the present Ms). Furthermore, the structures of these variants were solved and the arginine binding process was investigated by ITC measurements reporting a binding constant to D1 of 3,6 uM-1.

The present work, as mentioned before, investigates a potential practical application of these scaffolds in arginine biosensors and to this end a fluorescent mutant is obtained: the conservative (and I would add obvious) replacement Phe76Trp, located in the binding pocket, is identified as promising candidate just by visual inspection of the previously reported structure, as the authors point out in page 3, line 88.

In my opinion it is not surprising that a single point conservative mutation in the binding pocket does not change either the structure or binding process significantly, as it is shown (Kd=3,4uM).

Author Response

Comments and Suggestions for Authors

Arginine detection is of great interest as alterations in its levels have been associated to a number of human diseases. Smaldone et al. have developed a molecular scaffold that could be used as a promising tool in arginine biosensors. Starting from a simplified version of the Arginine Binding Protein from Thermotoga maritima (TmArgBP) and based on MD simulations, two fluorescent variants are obtained and characterized in terms of 3D structure and fluorescence properties upon arginine binding. Although the experimental work is well done from a methodological point of view, in my opinion it is not enough to reach the caliber of ijms:

In their 2018 International Journal of Biological Macromolecules paper (“Domain communication in Thermotoga maritima Arginine Binding Protein unraveled through protein dissection” 119, 758-709), the authors described the dissection of TmArgBP into its monomeric version and its D1 and D2 domains (Fig 16 in this paper is basically the same as Figure 1 in the present Ms). Furthermore, the structures of these variants were solved and the arginine binding process was investigated by ITC measurements reporting a binding constant to D1 of 3,6 uM-1.

The present work, as mentioned before, investigates a potential practical application of these scaffolds in arginine biosensors and to this end a fluorescent mutant is obtained: the conservative (and I would add obvious) replacement Phe76Trp, located in the binding pocket, is identified as promising candidate just by visual inspection of the previously reported structure, as the authors point out in page 3, line 88.

In my opinion it is not surprising that a single point conservative mutation in the binding pocket does not change either the structure or binding process significantly, as it is shown (Kd=3,4uM).

Response

We thank the reviewer for evaluating the manuscript and for the positive evaluation of the methodological aspects of the manuscript. Although we respect the legitimate opinion of the reviewer, we believe that the results described in the manuscript are of interest.

Although the identification of a residue of the active site as a candidate for the mutagenesis may appear obvious, there is in general no guarantee that the mutants possess the properties (stability, affinity for the ligand, and variation of the fluorescence upon binding) we were looking for. In particular in a case like this one where a Phe residue was replaced with a bulkier Trp residue. An analysis of the close contacts that the mutated residue makes with the ligand (see Figure S3B of the revised version of the manuscript) provides an idea on how critical could have been the replacement for the ligand affinity. It is worth mentioning that the inspection of the site was used only for the preliminary identification of the mutation site. Indeed, as it is evident from the manuscript, the experimental work was preceded by extensive molecular modelling and dynamics that supported the feasibility of the study.

In this context, it is important to note that such a mutation, independently on how obvious it was to be designed, had not been considered in the previous studies aimed at developing arginine biosensors from TmArgBP (see for example Deacon, et al. Biochimie 2014, 99, 208-214; or Donaldson, et al. Anal Biochem 2017, 99, 208-214).

Of course, we exploited the knowledge we acquired over the years on the possibility to extensively manipulate this protein. This prompted us to graft the mutation on an extremely simplified version of the protein constituted by a single domain (domain 1). In this way, we generated a scaffold that, in contrast to what is observed for TmArgBP and for other substrate binding proteins, could easily release the ligand. Taking into account the increasing interest for substrate binding proteins as metabolite sensors, we think that our findings open new design strategies in the field.

Round 2

Reviewer 1 Report

Authors performed all the changes suggested in the previous revision. Therefore, I recommend it for publication.

Author Response

Comments and Suggestions for Authors

Authors performed all the changes suggested in the previous revision. Therefore, I recommend it for publication.

Response

We thank the reviewer for the supportive evaluation of the manuscript

Reviewer 2 Report

I appreciate the authors reply. In any case I suggested that the results are not interesting. I still believe, nevertheless, that the results are scarce for a publication in IJMS and would be the basis for further investigation. Of course, there is no 100% guarantee of anything when doing protein engineering but, again, the Phe to Trp mutation in a relatively accessible site of the protein is a highly conservative substitution and I expect no big surprises.

I totally understand and appreciate the fact that the authors exploit  previous results on this protein to pursue practical applications, as it is the case. However, sentences in the Abstract as:

 “Here, we developed protein scaffolds suitable for Arg sensing through the dissection of the multimeric …”

and also

“Indeed, TmArgBP domain-swapped structure was manipulated to generate simplified monomeric and single domain scaffolds

are misleading. The only manipulation made on the protein, in this particular work, has been a single conservative mutation. It should be clear, in the abstract, what is new and what is not (protein scaffolds).

In legend to figure 2, panel (E) is missed.  

Author Response

Comments and Suggestions for Authors

I appreciate the authors reply. In any case I suggested that the results are not interesting. I still believe, nevertheless, that the results are scarce for a publication in IJMS and would be the basis for further investigation. Of course, there is no 100% guarantee of anything when doing protein engineering but, again, the Phe to Trp mutation in a relatively accessible site of the protein is a highly conservative substitution and I expect no big surprises.

I totally understand and appreciate the fact that the authors exploit  previous results on this protein to pursue practical applications, as it is the case. However, sentences in the Abstract as:

 “Here, we developed protein scaffolds suitable for Arg sensing through the dissection of the multimeric …” and also “Indeed, TmArgBP domain-swapped structure was manipulated to generate simplified monomeric and single domain scaffolds”

are misleading.

The only manipulation made on the protein, in this particular work, has been a single conservative mutation. It should be clear, in the abstract, what is new and what is not (protein scaffolds).

In legend to figure 2, panel (E) is missed. 

Response

Taking into account the observation of the reviewer, we changed the two sentences of the abstract as follows:

  • Lines 15-17: “Here, we developed protein variants suitable for arginine sensing by mutating and dissecting the multimeric and multidomain structure of Thermotoga maritima Arginine Binding Protein (TmArgBP).”
  • Lines 17-18: Indeed, previous studies have shown that TmArgBP domain-swapped structure can be manipulated to generate simplified monomeric and single domain scaffolds.

We amended the legend of figure 2. We thank the reviewer for pointing out that a reference to the panel E was missed in the legend.